# Self-Medication Practice and Associated Factors Among Health Professionals in Spain

**DOI:** 10.3390/nursrep15020053

**Published:** 2025-02-04

**Authors:** Eva Manuela Cotobal-Calvo, Concepción Mata-Pérez, Anna Bocchino, Ester Gilart, Belén Gutiérrez-Baena, José Luis Palazón-Fernández

**Affiliations:** 1Salus Infirmorum Nursing Center, University of Cádiz, 11001 Cádiz, Spain; evamanuela.cotobal@ca.uca.es (E.M.C.-C.); conchi.mata@ca.uca.es (C.M.-P.); belen.gutierrezbaena@ca.uca.es (B.G.-B.); jluis.palazon@uca.es (J.L.P.-F.); 2Department of Nursing and Physiotherapy, University of Cádiz, 11009 Cádiz, Spain; ester.gilart@uca.es

**Keywords:** self-medication, health professionals, attitude, perceived efficacy

## Abstract

(1) **Background**: Self-medication, defined as the use of medications without professional supervision, is a common practice that presents both potential benefits and significant risks. This study analyzes the prevalence, patterns, and determinants of self-medication among health professionals in Spain. (2) **Methods**: A cross-sectional descriptive design was employed with 438 health professionals, predominantly women (81.1%), with a median age of 42 years. The majority of the healthcare workers were nurses (45%). (3) **Results**: The results revealed a high prevalence of self-medication (59.4%). Analgesics and anti-inflammatory drugs were the most commonly used. Age and professional knowledge emerged as significant factors influencing this behavior. The main reasons for self-medication included the mildness of symptoms, easy access to medications, and previous successful experiences. Digital sources, especially websites, were the most consulted. Confidence in artificial intelligence tools as a clinical resource was moderate, with 18% of participants consulting AI tools, a rate comparable to the 19.5% for scientific databases. Logistic regression analysis identified age, knowledge of recommended doses, and perceived efficacy as significant predictors, while concern about risks acted as a protective factor. (4) **Conclusions**: This study highlights the need for educational interventions aimed at promoting responsible self-medication practices and mitigating associated risks among healthcare professionals.

## 1. Introduction

Self-medication is a fairly common practice, involving, as defined by the World Health Organisation (WHO), the use of medicines without professional supervision, often to treat minor ailments. Responsible self-medication involves the use of approved medicines available without prescription, which are safe and effective when used as directed [1,2]. Recent findings suggest that approximately 22% of people who use medications self-medicate, and notable increases have been observed over the years among different demographic groups [3].

While this behavior results in improved individual health by providing rapid relief of symptoms and/or clinical manifestations, it also presents significant risks, such as misdiagnosis, antimicrobial resistance, and adverse effects due to incorrect use of medicines [4,5]. In addition, frequent misuse of some medicines (i.e., anxiolytics) can lead to serious dependence and abuse problems, with both short- and long-term detrimental health consequences [6,7].

Previous studies have indicated that self-medication may be influenced by factors such as gender [8], knowledge [9] and easy access to medicines [10], inadequate prescription control [11], familiarity with treatment options and/or previous successful experiences with medicines, and lack of time [12].

The practice of self-medication has been particularly exacerbated by easy access to online health resources. Through the dissemination of a wide volume of medical data through the internet, digital technologies and artificial intelligence (AI) have enabled both doctors and patients to access information about medicines, diseases, and treatments quickly and efficiently [6,13].

Self-medication among health professionals raises relevant concerns. On the one hand, the possibility of health professionals using medicines without specific clinical guidelines may result in risks to their health, such as drug interactions or unexpected adverse effects [4], on the other hand, the search for information on unregulated platforms could lead to a false belief about the effectiveness and safety of certain treatments. Websites that are not reviewed by competent professionals may spread inaccurate information, which can lead to pernicious decisions based on unscientific data [14].

It is essential to promote a culture of education and responsibility in this context. Healthcare professionals must be critical and discern the quality of the information they consume, using reliable sources and consulting scientific evidence, prioritizing their continuous training [15].

In the scientific literature, there are several questionnaires designed to assess the practice of self-medication, which have been used both in the general population and in healthcare professionals. However, it is important to note that most of these instruments have not been updated to incorporate recent technological advances among the causes or reasons for such practices and/or in clinical decision-making. To date, only one questionnaire has been specifically adapted to reflect these advances, with the aim of assessing self-medication in healthcare professionals in a comprehensive manner, including the degree of trust that healthcare professionals place in clinical assessments generated by artificial intelligence tools [16].

Results based on the administration of this questionnaire could provide valuable information that could guide future intervention strategies and training programs aimed at improving the practice of self-medication in health professionals.

Therefore, the aim of this study is to analyze the prevalence, patterns, and causes of self-medication, indicating the most commonly used medicines, sources of information, and access to medicines, as well as knowledge and attitudes towards self-medication among health professionals.

## 2. Materials and Methods

### 2.1. Study Design and Participants

This descriptive cross-sectional study was conducted in a sample of health professionals, chosen by convenience sampling and recruited through their work environments and professional social networks (LinkedIn, Facebook, and WhatsApp groups). A link to the online questionnaire was provided to improve access and ensure greater participation. The sample included a wide range of health professionals, with the aim of analyzing practices, attitudes, and knowledge associated with self-medication. Using the Epidat 3.1 program, and a previous estimate of 82%, calculated by a pilot sample, a minimum sample size of n = 227 was calculated for a confidence level of 95% and a margin of error of 5%. Nevertheless, an attempt was made to include as many participants as possible.

### 2.2. Instruments

Data were collected through an online questionnaire of 26 items, created especially to evaluate aspects related to self-medication. The items were divided into the following sections:(a)Sociodemographic aspects: age, gender, profession.(b)Self-medication practice: prevalence, frequency, duration, reasons for self-medication diseases/symptoms treated, types of drugs used, sources of information about medications and pathologies, and adverse effects.(c)Knowledge about medication: degree of expertise about dosage, drug interactions, and related dangers(d)Perception of the effectiveness of and attitudes toward self-medication.

The items in Group B, which assessed self-medication practices, were completed only by healthcare professionals who declared they practiced self-medication.

The items in Groups C and D, which evaluated knowledge about medication (e.g., degree of expertise regarding dosage, drug interactions, and related dangers) and perception of the effectiveness and attitudes toward self-medication, were answered using a 5-point Likert scale.

The questionnaire was initially validated through a pilot study with 123 health professionals to ensure the clarity of the questions and their ability to obtain relevant information. The validation process included assessment of the clarity of the questions, their relevance to the study objectives, and their ability to capture the necessary data. Feedback was collected from participants during the pilot phase and adjustments were made accordingly to improve the reliability of the instrument [16].

### 2.3. Procedure

Participants were contacted through their telephones, emails, and/or social platforms. Initially, e-mails were sent to the coordinators of various hospitals and primary healthcare centers to distribute the information to other professionals. The questionnaire was administered online using Google Forms, which facilitated data collection. Prior to completing the questionnaire, those who agreed to participate were informed of the objectives of the study and gave their informed consent digitally, ensuring anonymity and the confidentiality of responses. The time to complete the questionnaire ranged from 10 to 20 min, with an average of 15 min.

This study adhered to the ethical principles outlined in the 2013 Declaration of Helsinki (7th revision, 64th meeting, Fortaleza) and complied with the Organic Law 3/2018 of December 5, regarding the protection of personal data and digital rights in Spain. Data collection was conducted with strict adherence to these principles, ensuring confidentiality, obtaining informed consent, and safeguarding the privacy of all participants.

In accordance with Royal Decree 1090/2015, which regulates clinical trials and Research Ethics Committees for Medicinal Products (CEIm), and Law 14/2007 on Biomedical Research, only research involving clinical interventions or the use of medical devices is required to undergo ethics committee review. Furthermore, the Regulation (EU) 2016/679 (GDPR) and Organic Law 3/2018 (LOPDGDD) establish that anonymized data are exempt from these regulations. Therefore, considering that this study posed no significant risks to participants and adhered to fundamental ethical principles, it was determined that formal review by an ethics committee was not required.

### 2.4. Statistical Analysis

Qualitative variables were described using absolute and relative frequencies. Quantitative and ordinal variables showed significant deviations from the Normal distribution (Kolmogorov–Smirnov test) and they were expressed as the median (Me) and interquartile range (IQR). Relationships between categorical variables were studied using the Chi-square test. Comparisons of quantitative variables between the two groups (self-medicating and non-self-medicating) were performed using the Mann–Whitney U test. Missing values were deleted in a pairwise fashion. All analyses were performed using IBM’s SPSS© (version 26.0) software for Windows (IBM Corp., Armonk, NY, USA). For all tests, *p*-values ≤ 0.05 were considered significant.

Using “backward” and “introduction” methods, multiple logistic regressions were adjusted to investigate the contribution of multiple independent variables: sociodemographic variables (age, sex, and profession), variables measuring knowledge, attitudes and the perception of risks, and the efficacy of self-medication practice in the prediction of self-medication. For those variables that were statistically significant, the corresponding odds ratios and their 95% confidence intervals were obtained.

## 3. Results

### 3.1. Sociodemographic Characteristics

The sample consisted of 438 participants (355 females, 82 males, and 1 person who preferred not to say their gender), with a median age of 42 years (IQR 25–52). The majority of the participants were nurses (45%), followed by intermediate and higher technicians and physicians (Table 1).

### 3.2. Practice of Self-Medication

#### 3.2.1. Prevalence of Self-Medication

Of those surveyed, 59.4% indicated that they self-medicated. There was no relationship between self-medication and sex (χ^2^ = 0.305; *p* = 0.581), but there was a relationship with age (U = 26,428; *p* = 0.007) and profession (χ^2^ = 24.97; *p* = 0.001). Those who self-medicate the most were physicians (81.3%), followed by nurses (65.0%). The rest of the health professionals also presented high percentages (above 40%). In terms of age, those who self-medicate were older (Me = 45 years; IQR: 27–52) than those who did not (Me = 30 years; IQR: 24–51.5).

#### 3.2.2. Frequency and Quantity of Medications Consumed by Self-Medication

The majority (71.5%) of the 260 participants who self-medicated indicated that they did so sporadically and only 4.6% that they self-medicated daily.

Regarding the number of medications consumed per week, 73.5% of the self-medicating respondents indicated that they consumed one medication per week, 16.9% consumed two medications per week, and 6.5% consumed three medications per week. In total, 96.9% took 1 to 3 medications per week. A small percentage (3.1%) indicated that they consumed four or more medications per week.

Concerning the question about when they stopped self-medicating, the majority (58.8%) indicated that they stopped taking the medications immediately after the symptoms disappeared, although this also depended on the medications consumed (33.5%) and/or the disease (23.8%) (Table 2).

#### 3.2.3. Reasons for Self-Medication

With regard to the reasons for self-medicating, the majority (67.3% of the 260 participants who self-medicated) indicated that it was due to the mildness of the symptoms, although about 40% indicated that it was due to easy access to the medications, previous experience with the medications, and having sufficient knowledge about the medications. It should be noted that a very small percentage (0.4%) indicated advertisements as the reason for self-medication, 4.6% were dissatisfied with the treatment prescribed by the physician, and 1.5% doubted the diagnosis made by the physician (Table 3).

#### 3.2.4. Sources of Information About Medications and Pathologies

In total, 209 out of 260 (80.4%) of the respondents who indicated that they self-medicated reported that they searched for information related to health/pathologies on the internet, AI, or social networks. Table 4 presents the frequency with which they used each of the various sources available, highlighting that the highest percentage corresponded to websites through the different search engines (61.0%), followed by health pages (39.0), the Spanish medications agency (37.1%), and health forums (32.7%). It is also important to highlight the low percentage of healthcare professionals who use artificial intelligence (18.0%), this being the least used source.

#### 3.2.5. Reasons for Seeking Information on Medications

The reasons for seeking information on medications/pathologies on the internet are presented in Table 5. It can be seen that the main reasons given were as follows: (a) to resolve doubts (50.2%), (b) to acquire or reinforce their knowledge as part of their professional development (40.5%), (c) to contrast information (39.0%), and (d) to seek information about the health problem (36.1%). In total, 32.2% indicated that they trusted the information found and 28.8% that they did so because of the immediacy provided by the internet to obtain the required information.

#### 3.2.6. Sources of Information Consulted by Health Professionals for Self-Medication

The majority of the participants who self-medicated (60.8%) indicated that before self-medicating they consulted a physician for information and 42.3% indicated that they had sufficient professional knowledge (Table 6).

#### 3.2.7. Illnesses Treated

Among the illnesses treated, a high percentage indicated that the main reason for self-medicating was to treat pain, followed by flu-like symptoms. The remaining symptoms/illnesses presented values of less than 25% (Table 7).

#### 3.2.8. Types of Medications Used in Self-Medication by Health Professionals

With regard to the types of medication, analgesics (80.8%) and anti-inflammatory medications (70.8%) were the most consumed (Table 8), both of which are available over-the-counter without the need for a prescription.

#### 3.2.9. Sources of Medicines for Self-Medication

Most of the participants who stated that they self-medicated (85%) acquired medicines or health products from pharmacies, 30.0% accessed these products through hospitals or their workplace, while 27.3% used the family medicine cabinet. Other less common sources included friends or family (10.4%), herbalists’ shops (6.9%), and online shopping (5.4%). The above figures are based on the 260 participants who stated that they self-medicated. Multiple answers were possible.

#### 3.2.10. Side Effects

Of those who responded that they self-medicated, 21 (8.1%) stated they had had some problems related to the medications used. The following table shows the problems in decreasing order of frequency, among which gastrointestinal problems stand out (Table 9).

### 3.3. Knowledge About Medications in Health Professionals

In general, the respondents indicated that they had a good knowledge of dosage, method of administration, adverse effects, and interactions. When comparing the scores on knowledge of the different aspects consulted between those who self-medicated and those who did not, differences were found in the distribution of the scores for all aspects, with the scores always being higher in those who self-medicated (Table 10).

### 3.4. Attitudes Towards Self-Medication in Health Professional

In general, the participants expressed a level of perception of the efficacy of self-medication in the range 2–4, which was significantly higher in those who self-medicated. In addition, they indicated that the availability and accessibility of over-the-counter medications influenced their decision to self-medicate. Those who self-medicated indicated that confidence in their knowledge about health and available treatments influenced their decision to self-medicate, with significantly higher values in those who self-medicated. Both groups indicated low levels of confidence in AI clinical advice and judgments about their health when self-medicating. Confidence in clinical judgments and opinions from internet discussion groups and forums was also low, although it was significantly higher in those who self-medicated. Regarding their level of concern about the risks associated with self-medication, both groups indicated high values, although this was significantly higher in those who did not self-medicate (Table 11).

### 3.5. Opinion on Self-Medication Compared to Medical Consultation to Treat Common Health Problems

Regarding their opinion on self-medication compared to consulting a physician, 54.6% indicated that they preferred medical consultation and only 3.7% that they preferred self-medication. A total of 33.8% of the participants indicated that depending on the situation, they preferred one or the other and 7.9% stated that they did not have a preference or had no opinion on this matter.

Finally, regarding the general opinion on the practice of self-medication, 37.1% indicated that it was useful in certain situations, 28.8% that it was acceptable in extremely mild situations, and 21.7% that it should be avoided as far as possible. Only 8.7% stated that it should always be avoided and 1.1% that it was always useful. The rest of the participants (2.6%) stated that they did not have a preference or had no opinion on this matter.

### 3.6. Logistic Regression Model for Self-Medication

The multivariate logistic regression showed the effects of each independent variable on self-medication, adjusted for the other. Table 12 shows the variables that had significant results as well as the adjusted odd ratios for each of them. The resultant logistic regression equation was as follows:Logit (P) = −2.202 + 0.020 × Age + 0.507 × KD + 0.610 × ES + 0.276 × RJG − 0.683 × RS

The model was significant (χ^2^ = 128.845, *p* = 0.000) and the Hosmer–Lemeshow test— χ^2^ = 9.202, *p* = 0.326—indicated a good fit to the data. In total, 80.7% of the respondents in the self-medication group and 60.5% in the non-self-medication group (overall 72.5%) were correctly classified by the fitted model. The final regression model explained self-medication in health professionals as a function of age, knowledge of recommended dosages, level of perception of the effectiveness of self-medication, reliance of clinical judgment of internet discussion groups and forums, and worries about risks of self-medication. The last variable acted as a protective factor against self-medication. The level of perception of the effectiveness and knowledge of recommended dosages had the greatest impact on the decision to self-medicate.

The logistic regression model showed that age, knowledge of recommended doses, and the perception of the efficacy of self-medication were significant predictors of this practice. These findings underscore the importance of these factors in self-medication decision-making. Furthermore, concern about the associated risks is observed to act as a protective factor, suggesting that increasing risk awareness could be key in future educational interventions. These results provide an important basis for strategies to encourage responsible medication use.

## 4. Discussion

The present study has revealed the complexity underlying the perceptions and attitudes of healthcare professionals towards self-medication. It also highlights the motivations that drive such practice and the emerging role of Information and Communication Technologies (ICTs) and AI in clinical decision-making. The main findings of our research are discussed below.

### 4.1. Gender and Self-Medication

In our study, no significant differences were found between genders regarding self-medication, contrary to the findings of other studies conducted with the general population [17,18,19]. However, it is worth noting the predominance of the female sex in the sample, likely due to their greater representation in health professions, particularly in nursing.

### 4.2. Age and Self-Medication

Regarding the relationship between age and the practice of self-medication, and in accordance with other studies conducted in different contexts [20,21,22], our results showed that this is more common in participants with a median age of 45 years. This finding could be attributed to a greater previous experience with medications and a greater self-confidence in their health-related knowledge.

### 4.3. Prevalence and Characteristics of Self-Medication

More than half of the participants claimed to practice self-medication, with a predominant frequency of 1 to 3 drugs per week. These results are consistent with the data obtained in a study on self-medication in university students [23]. In addition, similar to other studies, although in a smaller proportion, some health professionals reported consuming four or more drugs per week, mainly to treat mild symptoms such as common pain and flu-like symptoms, with the most commonly consumed groups of drugs being analgesics and anti-inflammatory drugs [23,24,25,26,27]. However, this practice is not without risk, since as other studies suggest [23,24,25,26,27] it can have adverse health consequences by significantly increasing the risk of drug–drug interactions and side effects such as gastrointestinal problems and liver damage [28,29,30,31]. However, in people who report self-medication, the perceived efficacy is quite high since, as previous studies indicate [32,33,34,35], it is related to an optimistic view regarding the results obtained.

### 4.4. Preference for Medical Consultation and Perceived Barriers

Despite the fact that the practice of self-medication is quite common, most of the participants report preferring medical consultation. However, factors such as delays in obtaining appointments and/or, in some cases, easy access to medications have increased this practice, especially among physicians and nurses who, as other studies indicate, are able to self-assess their health needs [36]. In fact, confidence in self-knowledge about health and available treatments also proved to be a key factor among participants.

### 4.5. Health Information Seeking and Perception of AI

Contrary to expectations, both groups (those who resort to self-medication and those who do not) reported low levels of trust in AI clinical advice and judgments regarding their health, which could reflect a distrust towards new technologies in health contexts, probably due to their recent implementation in the Spanish context.

In relation to the sources of information, the results indicated that health professionals continue to prioritize physicians and pharmacists to resolve doubts regarding the treatment and/or medications used. However, due to the digital boom, many people use the internet to acquire or reinforce knowledge or to contrast information, with a clear preference for generic websites and health pages, a result consistent with a previous study [36].

On the other hand, it is important to note that against expectations, and despite their relevance and credibility, the use of scientific databases (PubMed, SciELO, etc.) is relatively low, which may be related to the additional effort required to search, filter, and understand scientific articles.

Finally, the multiple logistic regression analysis on self-medication reveals that this practice is positively influenced by the perception of effectiveness of self-medication, knowledge of recommended doses, age, and trust in opinions from forums and online discussion groups. On the other hand, concern about the risks associated with self-medication acts as a protective factor. These findings are consistent with previous studies pointing to the importance of knowledge and perceived efficacy as key determinants of self-medication decisions [37,38].

In addition, the model presented an adequate fit, according to the Hosmer–Lemeshow test, which reinforces the validity of the results obtained.

These findings highlight the importance of designing interventions aimed at improving practical knowledge about dosing and risks among healthcare professionals. Furthermore, the protective role of risk concern suggests that educational strategies should include activities that promote critical reflection on the potential consequences of self-medication. Such interventions would not only contribute to reducing the associated risks but also foster a culture of responsible self-medication within the healthcare setting. By addressing these critical factors, healthcare organizations could mitigate potential harm while enhancing the overall safety and efficacy of self-medication practices among professionals.

This study has several limitations that should be considered when interpreting the results. First, it is a cross-sectional study, which limits the ability to establish causal relationships between variables. In addition, this design captures data at a single point in time, which precludes the observation of changes or trends over time that could provide a deeper understanding of the dynamics of self-medication. A randomized or longitudinal design would have been more appropriate for obtaining more generalizable conclusions and exploring causal relationships with greater precision. This limitation will be taken into account in future research, which may adopt more robust designs to further deepen the findings.

Second, the results are based on self-reported data, which introduces the possibility of recall or social desirability biases. These factors may underestimate the incidence, frequency, and possible consequences associated with self-medication, thus partially compromising the accuracy of the conclusions.

Moreover, the questionnaire used was designed and adapted for a specific population, which may limit its generalizability to other populations or contexts.

Lastly, this study used convenience sampling, which inherently introduces the possibility of selection bias. Efforts were made to mitigate these biases during the analysis phase, taking into account demographic variables such as age, sex, and profession, to minimize potential imbalances. While convenience sampling limits the generalizability of the findings, it was a practical approach, given the scope and objectives of the study. This limitation will be addressed in future research, which could adopt more robust sampling methods to improve the generalizability of the results.

## 5. Conclusions

The present study provides relevant information on the prevalence and factors associated with self-medication among health professionals in Spain, highlighting the relationship of this practice with the perception of efficacy, knowledge about dosage, and trust in information sources.

Although self-medication may offer immediate solutions for mild symptoms, our findings highlight the potential dangers that may derive from this practice, such as side effects and potential drug interactions. These results underline the importance of educational actions aimed at promoting a conscious use of drugs and minimizing the associated dangers.

However, these findings need to be interpreted with caution due to the methodological constraints of the study.

In future studies, it would be valuable to adopt longitudinal designs and more representative sampling methods to explore the dynamics of self-medication in greater depth and to evaluate the efficacy of the proposed interventions.

In conclusion, while this study contributes to knowledge about self-medication in the healthcare setting, further research is needed to strengthen the evidence base and guide specific strategies in this context.

## Figures and Tables

**Table 1 nursrep-15-00053-t001:** Distribution of participants by profession.

Profession	n	Proportion (%)
Nurse	197	45.0
Intermediate Technician	56	12.8
Higher Technician	51	11.6
Physician	48	11
Psychologist	25	5.7
Physiotherapist	23	5.3
Pharmacist	17	3.9
Biologist	10	2.3
Auxiliary Nursing Care Technician	3	0.7
Chriropodist	3	0.7
Dentist	3	0.7
Veterinarian	2	0.5

**Table 2 nursrep-15-00053-t002:** Duration of self-medication practice in health professionals. Multiple answers were possible.

Duration of Self-Medication	Proportion (%)
Immediately after symptoms disappear	58.8
Depends on the medications	33.5
Depends on the disease	23.8
Some days after the symptoms disappear	10.0
I take it for the long term	5.0
When the medications is depleted	1.2
Within a few days, regardless of the outcome	1.2

**Table 3 nursrep-15-00053-t003:** Reasons for self-medication in health professionals. Multiple answers were possible.

Reasons for Self-Medication	Proportion (%)
Mildness of symptoms	67.3
Easy accessibility	41.2
Previous experience with the drug	40.8
Sufficient knowledge about medicines	40.4
Delay in obtaining an appointment at the health center	30.4
Lack of time to attend a doctor’s appointment	17.7
Recommendation from family and friends	10.8
Confidence in the websites or forums consulted	5.8
Confidence in AI tools consulted	5.4
Dissatisfaction with the treatment prescribed by the physician	4.6
Being embarrassed to discuss the symptoms	1.9
Distrust in medical diagnosis	1.5
Advertisement	0.4

**Table 4 nursrep-15-00053-t004:** Digital sources consulted to obtain information about health and diseases by health professionals. Multiple answers were possible.

Resources	Proportion (%)
Search engines (Google, Bing, Internet Explorer, etc.)	61.0
Health pages	39.0
Spanish Agency of Medicines and Medical Devices	37.1
Health forums	32.7
Scientific articles	25.4
Drug applications	24.4
Social networks (Facebook, Instagram, etc.)	20.5
Websites of official health organizations	20.0
Scientific databases (Pubmed, Scielo, etc.)	19.5
Artificial Intelligence (Chat GPT, Bard, You.com, etc.)	18.0

**Table 5 nursrep-15-00053-t005:** Reasons for seeking information on medications/pathologies on the internet. Percentages are based on the 260 respondents who self-medicated. Multiple answers were possible.

Reasons	Proportion (%)
To resolve any doubts I may have in this regard	50.2
To acquire or reinforce knowledge as part of my professional development.	40.5
To contrast information	39.0
To seek information about the health problem	36.1
I trust the information I find	32.2
Immediacy	28.8
In every group or web page there are very well trained professionals	14.1
I prefer the ease of communication provided by the internet instead of going to a health centre.	11.7
I prefer to consult information with other people who suffer from the same pathology.	6.8

**Table 6 nursrep-15-00053-t006:** Sources of information for self-medication.

Source	Proportion (%) *
Physician	60.8
Professional Expertise	42.3
Pharmacist	24.6
Medications leaflet	23.1
Websites	22.3
Doctor or other health professionals found through internet search	16.9
Family member, neighbour or friend	12.3
Artificial intelligence	10.0
Social networking sites	9.2
Medical forums	4.6
None	4.2
Books	2.3
Vademecum	0.8
Advertisement	0.8
Nurse	0.4

* Percentages based on the 260 participants who stated that they self-medicated. Multiple answers were possible.

**Table 7 nursrep-15-00053-t007:** Diseases treated through self-medication by health professionals.

Symptom/Disease	Proportion (%) *
Aches and pains (headache, menstrual, stomach, joint, low back pain, etc.)	85.8
Flu-like symptoms, colds, fever, coughs	61.2
Heartburn, gastritis, digestive disorders	24.6
Allergies	21.5
Insect bites	13.8
Hangover	12.3
Infections, inflammations	11.5
Psychological problems (insomnia, anxiety, stress, etc.)	7.7
Burns	7.7
Pregnancy prevention/contraception	5.0
Obesity or overweight	3.1
Lack of appetite, decay, exhaustion	2.7
Increase muscle mass	1.9
Skin problems	1.0
Hyperlipemia	0.5
Hypertension	0.5

* Percentages based on the 260 participants who stated that they self-medicated. Multiple answers were possible.

**Table 8 nursrep-15-00053-t008:** Proportions of different medications used by health professionals. percentages based on the 260 participants who stated that they self-medicated. Multiple answers were possible.

Type of Drug	Proportion (%)
Analgesics	80.8
Anti-inflammatory	70.8
Stomach protector or antacids	26.5
Antihistamines	22.3
Topical creams	19.6
Vitamins	14.6
Natural drugs	13.5
Anxiolytics	6.2
Contraceptives	5.8
Corticosteroids	4.6
Antibiotics	4.6
Sedatives or hypnotics	1.9
Antidepressants	1.2
Antihypertensives	0.8
Laxatives	0.8
Retinoids	0.4
Lipid lowering agents	0.4
Muscle relaxants	0.4

**Table 9 nursrep-15-00053-t009:** Side effects of the practice of self-medication. Multiple answers were possible.

Side Effect	N (%)
Gastrointestinal problems	13 (61.9)
Drowsiness	3 (14.3)
Dizziness	2 (9.5)
Dry mouth	1 (4.8)
Rash	1 (4.8)
Masking of the acute illness	1 (4.8)
Allergy	1 (4.8)
Hormonal disturbances	1 (4.8)
Low blood pressure	1 (4.8)
Addiction	1 (4.8)

**Table 10 nursrep-15-00053-t010:** Knowledge of dosage, method of administration, adverse effects, and interactions of medications in health professionals.

Knowledge of…	Total Sample ^†^ n = 438	Do Not Self-Medicate ^†^ n = 178	Self- Medicate ^†^ n = 260	U	*p*-Value
Adverse effects of the medications taken	4 (3–4)	3 (3–4)	4 (3–4)	27,326.0	0.001 *
Recommended dosages	4 (3–5)	4 (3–5)	4 (4–5)	28,261.0	0.000 *
Possible interactions with other medications, food, and/or medicinal plants.	3 (3–4)	3 (2–4)	4 (3–4)	28,499.5	0.000 *
Possible interactions with drug and/or alcohol consumption	4 (3–5)	4 (3–4)	4 (3–5)	26,719.0	0.004 *
Method of administration	4 (3–5)	4 (2–4)	4 (3–5)	28,241.0	0.000 *

^†^ Figures are medians and interquartile ranges (in parentheses); U: Mann–Whitney U test; *: significant difference.

**Table 11 nursrep-15-00053-t011:** Attitudes towards self-medication in health professionals who self-medicate and do not.

Variable	Total Simple ^†^ n = 438	Do Not Self-Medicate ^†^ n = 178	Self- Medicate ^†^ n = 260	U	*p*-Value
Level of perception of the effectiveness of self-medication	3 (2–4)	3 (2–3)	4 (3–4)	33,204.5	0.000 *
The availability and accessibility of over-the-counter medications influences your decision to self-medicate	4 (2–4)	4 (2–4)	4 (3–5)	23,469.5	0.795
To what extent confidence in your own knowledge about your health and available treatments influences your decision to self-medicate rather than consult a physician	4 (3–5)	4 (3–4)	4 (4–5)	28,899.0	0.000 *
To what extent does confidence in the AI’s clinical advice and judgments about your health influence your decision to self-medicate rather than to consult a physician	2 (1–3)	1 (1–3)	2 (1–3)	24,558.5	0.243
To what extent reliance on clinical judgments and opinions from internet discussion groups and forums about your health influence your decision to self-medicate rather than consulting a physician	2 (1–3)	1 (1–3)	2 (1–3)	28,178.5	0.000 *
How worrisome you consider the risks associated with self-medication	4 (3–5)	4 (4–5)	4 (3–4)	14,319.5	0.000 *

^†^ Figures are medians and interquartile ranges (in parentheses); U: Mann–Whitney U test; *: significant difference.

**Table 12 nursrep-15-00053-t012:** Stepwise logistic regression results for self-medication among health professionals.

Variables	Coefficient (β)	S.E.	Wald	*p*-Value	OR	95% CI for OR
Age	0.020	0.008	6.310	0.012	1.020	1.004–1.036
Knowledge of recommended dosages (KD)	0.507	0.120	17.951	0.000	1.661	1.313–2.100
Level of perception of the effectiveness of self-medication (ES)	0.610	0.112	29.525	0.000	1.841	1.477–2.295
Extent to which reliance on clinical judgments and opinions from internet discussion groups and forums about your health influence your decision to self-medicate rather than consulting a physician (RJG)	0.276	0.093	8.718	0.003	1.318	1.097–1.583
How worrisome you consider the risks associated with self-medication (RS)	−0.683	0.133	26.321	0.000	0.505	0.389–0.656
Constant	−2.202	0.850	6.707	0.010	0.111	

Note: only significant coefficients are shown. S.E.: standard error of coefficient; OR: odds ratio; CI: confidence interval. Abbreviations of variable names in brackets are the codes used in the regression equation.

## Data Availability

The data presented in this study are available on request from the corresponding author. The data are not publicly available due to privacy.

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
