# Peer review of "Self-Medication Practice and Associated Factors Among Health Professionals in Spain"

_nursrep, 2025, doi:10.3390/nursrep15020053_

Round 1
Reviewer 1 Report
Comments and Suggestions for Authors
Introduction: While the introduction provides a general background, it lacks a comprehensive review of recent literature and does not fully address the unique challenges posed by self-medication in healthcare professionals. Including more relevant and updated references would strengthen this section.
Research Design: The design is appropriate for a preliminary exploration of the topic but lacks the depth needed to provide generalizable conclusions. A randomized or longitudinal design would be more effective.
Methods: The methods section is insufficiently detailed. For example, it does not describe the process of questionnaire validation in depth or discuss how biases introduced by convenience sampling were addressed.
Results: The results are generally clear but could benefit from a more robust discussion of statistical significance and implications for practice. Visual aids such as figures and tables are helpful but need better contextual explanation.
Conclusions: The conclusions do not sufficiently align with the presented results. They are overly broad and speculative, given the limitations in the methodology and analysis.
Quality of English: While the English is generally comprehensible, it could be improved for clarity and precision, particularly in the results and discussion sections. Sentences are occasionally overly complex, and some technical terms lack proper explanation.
Moreover:
Abstract:
"Analgesics and anti-inflammatory drugs being the most commonly used."
This is a fragment and should be rephrased for clarity (e.g., "Analgesics and anti-inflammatory drugs were the most commonly used.").
Introduction:
"Respon- sible self-medication..."
The word "Responsible" is split incorrectly across lines.
Methods:
"Us- ing the Epidat 3.1 pro- gram..."
The word "Using" and "program" are improperly split.
Figures and Tables:
Some figure captions are inconsistent in formatting or have awkward phrasing. For example:
"Figure 5. Proportion of use of the different medicaments by health professionals."
The phrase "Proportion of use of the different medicaments" could be more concise, e.g., "Proportions of Different Medications Used by Health Professionals."
General Formatting:
Several hyphenated words break across lines (e.g., "medi- cations," "inter- ventions"), which affects readability.
Author Response
Dear Reviewer,
Thank you very much for your valuable comments. We sincerely appreciate the time and effort you dedicated to reviewing our manuscript.
Attached, you will find the revised document with all the corrections marked in red. We hope this approach facilitates reviewing the changes we have made in response to your feedback.
Please let us know if you have any further questions or require additional clarifications.

Reviewer 2 Report
Comments and Suggestions for Authors
Peer-Review Comments_01.08.25
Brief Summary:
This study examines self-medication use among health professionals in Spain utilizing a validated questionnaire. This study represents an updated view of self-medication as the questionnaire includes updated information on the impact and use of new technologies, such as artificial intelligence. The study emphasizes the high prevalence of self-medication among health professionals and differences in factors, such as age, knowledge of recommended doses, and perceived efficacy as predictors of self-medication use. This study can be utilized for basis of future direction for research studies and implementation of educational efforts for healthcare professionals on self-medication.
Overall, a very interesting and timely study providing very detailed information on the practices of self-medication use among healthcare professionals in Spain.
General and Specific Comments:
General comments:
- Could consider changing the term medicament to medications or if want to be more specific could utilize over-the-counter medications and dietary supplements, etc.
- Could consider adding the sample size (n) that was included for creation of the figures for additional clarity. For example, for Figure 3 – in the text it is stated that 80.4% of the respondents reported on searching for information on health on the internet; may consider (either in the text or figures) to add that the figure is looking at 352/438 respondents.
o Also, may consider adding the n to the text, especially when only looking at one group (e.g., in line 146 only referring to 71.5% of the self-medicated respondents).
- May consider making clearer which questions participants were allowed to select more than one response. For example, in Table 2, Figure 2, etc. (based on the percentages), participants could select more than one answer for source of information for self-medication (>100%).
- Use a consistent terminology for the participants – some use of health professionals vs. health care workers.
- Consistent use of ‘,’ vs. ‘.’ when using as decimal place. Much of the figures have ‘,’ and the text has ‘.’
Abstract:
- Could considering adding that the majority of the healthcare workers were nurses (45%).
- Line 17 – could consider specifying the type of websites (if available), i.e., if they were healthcare professional-focused websites. From Figure 3, it appears that webpages were the most frequently consulted (61%). Is this the result that you are referring to? If so, please update to reflect same terminology as Figure 3. Also, could consider changing this to search engine instead of webpages to more accurately describe what is in parenthesis (Google, Bing, Internet explorer, etc.).
- Would consider re-phrasing the statement about AI here. Your results have a few data points on AI – this statement in the abstract (lines 17-18) seems to be referring to the ‘confidence in AI tools’ from Figure 2. Based on Figure 3, AI had a similar consult number at 18% compared to 19.5% to scientific databases; although, this is technically lower than the other digital sources consulted, this number being very close to scientific databases is meaningful as your study is highlighting these newer resources available. Can consider clarifying that this is referring to the confidence in AI, not necessarily utilization.
Introduction:
- Line 26: Spell out WHO abbreviation.
- Line 35: Consider replacing ‘some medicines’ with specific medication examples referenced in the article.
- Lines 46-50: Is there a high prevalence in other studies (or article cited) of healthcare workers utilizing unregulated platforms or not accessing clinical guidelines? Would these healthcare professionals not have an increased likelihood of having access to professional resources/platforms?
- Line 65: Could expand on this (add a sentence or two) to make clearer that this study is an application of this previously developed questionnaire.
Materials and Methods:
- How did you ensure the individuals were healthcare professionals? Was there a question about their associated employer/position? What was your inclusion criteria for what qualified as a healthcare professional?
- Could you expand a little more on the pilot sample by a sentence or two? Was this part of a previously published work where the estimate was calculated or a pilot sample conducted prior to initiation of this study in order to determine the estimates? It appears that it likely was part of the citation 15 – could consider combining this section with lines 97-99 to make clear that the estimates and validation of the questionnaire is coming from the pilot study reference 15.
- Lines 83-85: End sentence at possible. Move this additional information to the results section.
- Could consider numbering or lettering the sections of the questionnaire for better visual appeal/comprehension.
- Could also consider adding a statement that participants answered all 26 item questions regardless of prior responses (i.e., was not an adaptive questionnaire, using skip logic).
- Lines 102-103: any numbers available on how many healthcare centers the questionnaires were sent via email? Also, this section states that this was done initially – was the switch to other contact methods based on a low response rate from these health care center emails?
- Lines 107-108: This information may be more relevant in the results section. Also, may be helpful to include range of time.
Results:
- Line 133: IQR available for median age?
- Can you expand more on the definition of a technician? Also, what defines a middle vs. high level technician?
- Line 144: Could consider defining Me (median) and adding an IQR.
- Line 150: Is this value meant to be >1 and ≤3? If a respondent was taking a medication once per week, they would fit into both categories (1 medication per week) and (1-3 medications per week).
- Line 168-170: Clarifying question: these internet resources were only consulted for health/pathologies/disease information and not information on the medications?
- Line 172 and Figure 3: Could you please expand on what ‘health pages’ are? If possible, give some examples.
- Lines 173-175: Similar to comment above in abstract section, important to highlight that although this number was the lowest of the categories, still was similar percentages to other categories.
- Line 183: 40.5%
- Line 183: Text states to ‘confirm’ information and Figure 4 states to ‘contrast’ information. Please clarify which is correct. Note, contrast is utilized in discussion (line 312).
- Section 3.2.8: Could consider adding additional information here about what medications from these categories are available by prescription only vs over-the-counter.
- Section 3.2.9: Similar to prior comments – need further clarification on these percentages and n. Are these percentages only from patients who self-medicate?
- Table 5: These values need to be clarified. Are these values in the ‘total sample’, ‘do not self-medication’, and ‘self-medicate’ columns representing a scale where respondents rated these items? If so, what scale was used and what did values represent? If not, what do these median values represent?
- Table 5: Would recommend using parenthesis here to represent () vs. bracket = []
- Line 216-219: Median scores often the same, but differ in IQR. Consider re-phrase.
- Line 225: How is “medium perception” defined?
- Table 6: Same comment as above – please clarify scoring methodology/tool utilized. And switching brackets to parenthesis.
- Lines 241-244: This represents 92.1%, what was the other option/category(s)?
- Lines 245-248: Similar comment to above: totals add to 97.4%, other category(s)?
- Line 254 and Table 7: abbreviation shown as RJG and RIG for (what appears) to be referring to the same variable. Please clarify.
- Table 7: note section – change brackets to parenthesis.
Discussion:
- Line 271: write out abbreviation of ICTs
- Limitations section (starting on line 328): could consider adding sentence about limitation of convenience sampling.
Author Response

(The authors gave the same response as above.)

Round 2
Reviewer 1 Report
Comments and Suggestions for Authors
I think the authors cleared all of my issues. Please make sure that the images are of higher quality and the letters are the same font as the letters in the paper, as well as the font size (the text in Figure 4 is barely visible).
Author Response
Dear Reviewer,
Thank you very much for taking the time to review our manuscript and for providing such valuable feedback. Following the recommendations of the academic editor, we have replaced all figures with tables to further enhance the article’s readability.
We greatly appreciate your input, which has been instrumental in improving the quality of our work.
Please let us know if there are any additional comments or adjustments required.
Best regards,
Anna Bocchino